# The Role of Chemical Composition of High-Manganese Cast Steels on Wear of Excavating Chain in Railway Shoulder Bed Ballast Cleaning Machine

**DOI:** 10.3390/ma14247794

**Published:** 2021-12-16

**Authors:** Janusz Krawczyk, Michał Bembenek, Jan Pawlik

**Affiliations:** 1Faculty of Metals Engineering and Industrial Computer Science, AGH University of Science and Technology, 30-059 Kraków, Poland; jkrawcz@agh.edu.pl; 2Faculty of Mechanical Engineering and Robotics, AGH University of Science and Technology, 30-059 Kraków, Poland; jan.pawlik@agh.edu.pl

**Keywords:** ballast cleaning, rail ballast, Hadfield cast steel, manganese cast steel, wear mechanism of cast steel, Mangalloys

## Abstract

The main task for a ballast bed is to transmit the sleeper pressure in a form of stress cone to the subsoil, provide proper drainage and resist the sleeper displacement. Poorly maintained ballast could severely limit the maximum speed capacity and create further problems with the structural integrity, possibly leading to a complete failure of a given rail line. To prevent the unwanted corollaries, the ballast bed has to be periodically cleaned with an appropriate machinery. In this paper the authors investigated the effect of the chemical composition on the physical properties of the ballast excavating chains made of high-manganese steels. The authors focused on the wear mechanism, work hardening ability and hardness in the cross-sections areas. A microstructure analysis was performed as well, and observations revealed divergent morphology of precipitations at the grain boundaries, which influenced the size of austenite grains. The deformation twins formed as a result of operation were noticed in the samples. Research has shown that less carbon and chromium reduces the hardness of cast steel, and it specifically affects the ability to strain hardening. The authors explained the role of adjustments in chemical composition in the operational properties of high-manganese cast steels. It has been shown in the paper that different chemical compositions affect the properties of the alloys, and this causes different types of wear. The high content of chromium increases the hardness of materials before and after plastic deformation hardening, which in the conditions of selector chains results in greater dimensional stability during wear of holes in pin joints and will be more susceptible to abrasive wear in the presence of dusts from the ballast than creep.

## 1. Introduction

The general obstacle in stability of any system is the continuous necessity to put much effort in maintenance, which happens to use the resources that could be otherwise used for development [1,2,3]. As the cost of maintenance grows, new maintaining subsystems appear to be cost-effective, yet likewise they are prone to another kind of wear [4].

The issues of transportation maintenance and damage prevention can be divided into two parts: the issues connected with the conservation of the infrastructure and the problems addressed to the particular dynamic mean of transport itself [5]. In the case of railway system, the maintenance of the infrastructure is a complex subject, responsible not only for safety of passengers and goods, but also for efficiency of the whole logistic system [6,7]. This maintenance necessity is connected with every sub-part of the rail track: rail profiles, fastenings, sleepers and the ballast bed (Figure 1)—each having its own methodology of wear evaluation [7,8]. While keeping the rail track in a good shape is obvious to any observer, he or she may omit the importance of bedding conservation.

The main task for a ballast bed is to transmit the sleeper pressure in a form of stress cone to the subsoil, provide proper drainage [9,10,11] and resist the sleeper displacement [12,13,14]. Poorly maintained ballast could limit the maximum speed capacity severely and create further problems with the structural integrity, possibly leading to a complete failure of a given rail line [15].

In order to behave as designed, it has to be resistant to the environmentally induced degradation processes, such as erosion or clogged rainwater culvert. To prevent the unwanted corollaries, the ballast bed has to be periodically cleaned by an appropriate machinery [16,17,18,19]. Such bed ballast cleaning machine—despite having a complex design features—can be reduced to following sub-structures: the drive, control unit (with operator), rail track guideway, excavating unit, ballast transport unit, screening unit and reclaimed ballast placing unit [20]. Every working part of spoken sub-structure has to be manufactured out of proper material, able to withstand the high loads and be resistant to metal-mineral abrasive wear. Those steels usually belong either to the carburizable or nitridable materials [21,22] or to the family of Hadfield (and other high-manganese steel alloys, often times named “Mangalloys”) [17,20,23], but some manufacturers perform experiments with other types of metal mixtures [24].

Typical excavating chain is equipped with scraper shovels, usually with two to four fingers, each of them resembling conical picks, known from mining industry [25]. However, in the mining industry, conical picks are considered as consumables, as they work under high loads and are aggressive metal-mineral abrasive wear [26]. In the railway maintenance they happen to withstand the working conditions at a satisfactory level, making the excavating chain links the most wearable part in the cleaning system [27], yet other researchers claim that there is also room for improvement in the subject of frame and body structure of the ballast cleaning machine [28].

There are many studies concerning the connection between wear resistance and chemical composition of the various Hadfield steels in railway industry, yet those investigations are usually related to the wear of railway crossings [29,30,31,32,33,34]. Where it comes to applications similar to the bed cleaning machinery, most of the published papers refer strictly to the use of Mangalloys in the mining and quarry industry [35,36,37,38,39], though some of the wearing mechanisms appear to differ.

The high-manganese cast steels present specific, surface self-hardening properties, however it appears only under high loads since the mechanism relies mostly on dislocation twinning [40,41]. In the particular application of ballast cleaning, the working elements are subjected mainly to two types of wear—erosive wear caused by granular materials and creep wear resulting from persistent mechanical stresses [20]**.**

The proper chemical composition of this special family of cast steels is crucial to obtain expected results. The general structure of Hadfield steels is composed of austenitic grains with complex M_3_C (namely Fe_x_Mn_y_C [42]) carbide precipitations within the grains and on the grain boundaries [43]. Manganese, along with nickel and cobalt are widely used as an austenite forming and stabilizing elements [44,45,46]. The admixture of manganese is not a carbide former itself, it rather increases carbon solubility in Mn-Fe alloys [47]. That allows cementite matrix to precipitate during cooling of the cast steel, and thus constrain the growth of austenite grains. Other chemical elements in the alloy are responsible inter alia for the size of the austenite crystals [48]. During the heat treatment part of the cementite melts and supersaturates the austenite with carbon and that can result in self-hardening ability of the material and in some cases even strengthen the whole structure of solidifying alloy due to internal stresses [49]. In properly cast example, the austenite grains under high loads can slide along one of many dislocation planes and manifest the twinning, which strengthens significantly the loaded region of the structure.

In this paper the authors would like to focus on the effect of the chemical composition on the wear of the aforementioned ballast excavating chains. The samples were provided by Plasser & Theurer company (Vienna, Austria), which specializes in manufacturing heavy machinery for railway industry. Two different samples sets from real-life applications with different alloy composition were tested metallurgically and the wear-caused deflections were measured. The authors focused on the wear mechanism, work hardening ability and hardness in the cross-sections areas. A microstructure analysis was performed. The result of said wear in this very case resembles effects of intentionally applied burnishing treatment [50]. The key aim of the article is to explain the mechanisms of wear of chains made of high-manganese alloys and their surface work-hardenability resulting from the chemical composition of the alloy.

## 2. Materials and Methods

To perform the measurements, the authors chose worn scraper shovels and the corresponding linkages, shown on Figure 2. Those shovels were manufactured with high-manganese cast steel, and originally were utilized in a working chain of RM 80 UHR bed ballast cleaning machine, fabricated by Plasser & Theurer (Vienna, Austria). Basing on the technical documentation the scraper shovel, its linkage and other connecting parts were reproduced as digital models in SolidWorks 2018 environment (Dassault Systemes, Vélizy-Villacoublay, France) (Figure 3). The manner of enumerating the particular samples, based on the type of the element is shown in the Table 1.

### 2.1. Chemical Composition Analysis

Chemical composition was identified with the Foundry-Master (WAS) optical emission spectrometer (Hitachi, Tokyo, Japan).

### 2.2. Geometric Measurements of the Magnitude of Wear

The measurements of the dimensional changes caused by wear mechanisms were carried out in the area of the peg connection existing between each segment of the excavating chain (Figure 4.). To conduct the geometric deviation assessment of the diameters and depth of craters at the abrasive region, the authors utilized a metrologically certified digital calliper MarCal 16 EWRi manufactured by Marh GmBH (Goettingen, Germany).

### 2.3. The Analysis of the Microstructure

In order to make microstructural analysis possible, the authors extracted cut-outs from the studied parts according to the plan shown on Figure 5. The samples from the scraper shovel and the connector chain linkage were cut and grinded in such a manner, that they had two smooth and parallel planes, as is shown on Figure 6. The samples were enclosed in a resin scaffold and polished. The polishing process included sanding with a sandpaper of different grits (120–1200 grit) and finishing polish performed on a Struers Pedemax-2 polishing machine (Struers, Ballerup, Denmark) with an Al_2_O_3_ suspension in order to obtain a mirror-like surface.

The prepared surfaces of the samples cross-sections were etched with 6% nital. The microstructure at such conditions was examined using a Carl Zeiss Axiovert 200 MAT microscope (Carl Zeiss Microscopy Deutschland GmbH, Oberkochen, Germany). Average grain size for samples SS1 and SS2 (unloaded) was determined using the Jeffries planimetric method with Saltykov rectangle correction.

### 2.4. Hardness Test

#### 2.4.1. Hardness before and after Strain Hardening

The study of hardness and deformation level consisted of two separate stages—initially the samples were subjected to Brinell hardness testing (HBW). The goal of this activity was to locally harden the particular steel sample as a consequence of dislocating the grains of the crystal structure. Subsequently, in the very same spot a Rockwell (hardness test (HRC) was performed in order to evaluate the work-hardening ability of the material. This activity was undertaken to simulate the working conditions of the excavating chain. This measurements were afterwards compared to the values of HRC hardness of the untreated steel region. Hardness measurements were made using Zwick/Roell ZHU 187.5 (Zwick Roell Group, Ulm, Germany) hardness tester. The authors consider this method of evaluation of surface hardenability of high-manganese cast steels as a novelty, as it was not mentioned in the found literature.

#### 2.4.2. Cross-Sectional Hardness Inspection

The tests were carried out using the FM-700 Hardness Tester (InnovaTest, Maastricht, The Netherlands). The measurement points were located every 0.5, 1, 1.5, 2, 2.5, 3, 4, 6, 9 and 13 mm from the edge of the peg hole. Each of the samples was subjected to 10 measurements. The total distance from the first to the last measurement point from the eroded contact edge was 13 mm. Hardness was measured using the Vickers method with a load of 4.90 N (HV0.5). The time of measurement was set to 10 s.

## 3. Results and Discussion

### 3.1. Chemical Composition Analysis

The outcome of the chemical composition analysis for inspected cast steel is visible in the Table 2. The manufacturer documentation states that the scraper shovels and connection linkages should be cast out of GX120Mn12 steel, yet the spectroscopy study shows that there are several discrepancies: increased chromium share along with the lack of molybdenum in the sample SS2 and lower chromium and carbon content in the specimen ChC1. It should be expected that lack of molybdenum will be a cause of carbide precipitation inside the austenite grains. Lower amount of carbon and chromium in composition of ChC1 may have a negative impact of achievable level of hardness. Other alloy substrates were kept within the anticipated limits.

### 3.2. Wear-Caused Dimensional Deviation Measurements

Measurements of the material wear were conducted by checking the dimensional deviation of the excavating chain peg holes with respect to the original size. According to the documentation supplied by the manufacturer, the nominal diameter of the peg hole is ϕ 34.2 mm. The results of the comparison are visible in Table 3.

The measurements show, that—given the operational principle of the parts—there are no significant differences in the dimensional deviation between the two inspected materials. One can assume that the loss of dimensional stability is comparable for every sample. However, comparing the wear and deformation of the holes along their entire length, it is clearly visible that the SS1 and ChC1 samples, compared to the SS2 and ChC2 samples, are characterized by a larger dispersion of diameter sizes along the length of the holes. The differences between wear of the scraper shovel peg holes and connecting chain linkage peg holes are a result of a by-design different thickness of the particular parts.

### 3.3. The Analysis of the Microstructure

The microstructure of the samples SS1 and SS2 are shown on the Figure 7. These are the base material structures with no visible twinning. The twinning are a result of a crystalline dislocations, caused by subjecting the material under high surface pressure.

In both SS1 and SS2 cases, a dendritic setup is visible in the microstructure. It is worth to underline, that for the SS1 (un-worked, base material) the dendrites are much distinctive and more ordered. The differences of the average austenite grain size are collected in the Table 4.

The size of austenitic grains in SS1 sample is considerably bigger than of those in SS2. The reason behind the grain size differences, distinctive dendritic segregation scheme and arm spacing might be either the method of shovel casting or the heat treatment (heating to supersaturation temperature). On the Figure 8 and Figure 9 the authors wanted to show the placement of the carbides, appearing in samples SS1 and SS2.

Comparing both samples one can observe that in the case of material used for sample SS1 the precipitations occur inside the austenitic grains, whereas in case of SS2 they tend to occur near the grain boundary. That observation explains the fine-grained structure of SS2, where the precipitations limit or block the growth of crystalline grain.

A close-up on the structure located nearby the contact working surface for both types of steel is presented on Figure 10 and Figure 11. As those are the most strained regions of the given Mangalloy steel bodies, expected disloaction twinnings are perceptible.

Wear marks as well as work hardening marks are occurring in both samples—as a microcracks and loss of material or dislocation twins respectively. The surface of the material (SS1) seems to be slightly more prone to deformation (hence more twinning), than the material used for a sampleSS2.

### 3.4. Hardness Test

#### 3.4.1. Study of Hardness before and after Strain Hardening

The conducted study allowed the authors to analyse the deformation hardness reinforcement. As previously stated, the Rockwell hardness measurements were taken for all samples, both in the “raw” regions (HRC_0) and regions reinforced by strain hardening via loading the surface with Brinell test indenter (HRC_1). The results are presented below, in the form of Table 5. The obtained results may indicate that samples of materials with higher hardness both before and after hardening; i.e., SS2 and ChC2, will be more prone to abrasive wear in the presence of dusts from the ballast, while samples SS1 and ChC1 will have a greater susceptibility to be worn by creep.

The relationship between the Rockwell hardness (before and after “Brinell reinforcement”) and the content of carbon, chromium and molybdenum ispresented on the Figure 12. The authors decided to focus only on that three elements, since those are the factors that are making the difference between chemical composition of the scrutinized alloys.

The lowest initial hardness (before the reinforcement) was observed in the sample ChC1, with the lowest content of carbon and chromium. The biggest effect of hardness reinforcement was noticed in samples SS1 and ChC1—in both cases the hardness level was increased by a 3.3 factor. Those samples contained respectively: 1.2% Cr, 1,15% Mo (SS1) and 1.13% Cr and 1.01% Mo (ChC1). The highest hardness overall was obtained in the sample with the highest share of Chromium (SS2) after simulated operational work hardening.

#### 3.4.2. Cross-Sectional Hardness Inspection

The measured values of Vickers microhardness for the analysed samples are presented in Table 6 and on the Figure 13. The test was performed by making diamond-shaped indents every 0.5 mm, starting from the contact working surface of the peg hole. This study shows that strain hardening phenomenon has occurred on the contact surface during the machine operation. This effect is connected with the dislocation twinning of the grain structure, what helps the part to remain in the desired shape.

## 4. Discussion

The overall conclusion of the study is that the materials, even considering the differences in the chemical composition, behave similarly in the case of operational wear. The microscopic observations of the crystalline structure shown, that the specimen SS1 and ChC1 contained higher amount of idiomorphic austenite grains, with carbide grains enclosed within. As for the SS2 and ChC2, the precipitations were visible mostly at the grain boundaries, which resulted in allotriomorphic growth of austenite grains. The main difference in the used materials appeared to be the austenitic grain size. The microstructure of the material indicates that some of the carbon and the alloying elements (chromium) remain bound as carbides. For this reason, the lower content of carbon and chromium in the austenitic matrix reduces its potential for deformation strengthening by micro-twinning. It should be assumed that the supersaturation process can be carried out with greater efficiency than in the analyzed cases. In this regard, it is important to increase the solution annealing temperature.

High-manganese cast steels have tendency to create dislocation twins in their microstructure in response to the operational loads. In both cases there is observable effect of strain hardening near the contact working surface; nevertheless, sample SS1 and ChC1 appear to have the twinning phenomenon more intense than the sample SS2 and ChC2. This means that material strengthening is an important parameter in the analyzed friction node. Erosive wear by low-energy particles plays a lesser role. Carbide morphology does not result in decohesion during the strengthening of the surface layer as a result of tribological interaction (strengthening by micro-twinning).

The study of hardness of given materials is shown, that the lowest hardness values were obtained by the steels with lowest content of carbon and chromium. Consequently, the highest hardness after structure reinforcement was presented by the material with highest chromium content. The biggest difference of pre- and post- reinforcement hardness emerged in the samples SS1 and ChC1. Coherently with the twinning, the microhardness test conveyed that the highest hardness of operating unit was achieved in the working surface zone of the excavating chain elements. The analysis of differences in hardness hints, that dislocation twinning reinforcement occurs at the time of performing the hardness test, thus material presents strain-hardened properties even outside the area of operational exploitation. The authors also claim that the range of the superficial hardness reinforcement achieved in the laboratory is within the range of achievable hardening in the real-life application; however, in the proximity of the working zone (approximately 2 mm) the phenomenon of micro-twinning is observably stronger in the working elements. The biggest micro-twinning effect can be noticed in the sample ChC2, whilst the weakest is seen in ChC1. The factor responsible for the differences lays in the chemical composition—there is high content of carbon, chromium and molybdenum in the ChC2, whereas the share of C and Cr in the sample ChC1 is relatively low.

Reducing the content of carbon, chromium and molybdenum reduces the hardenability of the material. Reducing the grain size by increasing the chromium content and reducing the molybdenum content has no significant effect on the exploitation wear of the material under the working conditions of the tool.

The findings presented above match with some of the study results of Lindroos et al. [39], despite that in their research only one type of Mangalloy was examined. Moreover, they also developed a technique of pre-straining the surface of Hadfield steel, yet their method was based on a dynamic projectile, what brought additional deflection mechanisms into the account. The admixture of chromium also seemed to decrease the potential ductility in the work of Tęcza et al. [38], where microstructure analysis was performed, despite of lack of any hardness test. Our findings also line up with the conclusions of Atabaki et al. [52]; nevertheless all of the similarly-conducted studies cited above are referring to other applications of Hadfield steel, namely jaw crushers, which work under different stress conditions.

## 5. Conclusions

Presented research utilizes an innovative approach for a quantitative estimation of the surface work-hardenability in high-manganese cast steels by consecutive application of Brinell and Rockwell hardness testers in the same spot.

The micro-hardness test proved that the alloys with the lowest amount of chromium and carbon have the lowest hardness, which was expected. Higher contents of chromium (SS2 and ChC2) resulted in higher hardness (both before and after straun reinforcement), what on one hand resulted in higher dimensional stability of the pin holes; nevertheless, the part was prone to micro-cutting and mass loss on the other. That leads to a conclusion, that the dominant cause of wear in high-chromium Mangalloys used in ballast cleaners is the abrasive wear, being a result of working in an environment of fine-grained, sharp dust particles. It can be assumed that in the case of cast steels with a lower chromium content (SS1 and ChC1), the mechanism of wear relies rather on plastic deformation and material flow of the excavation chain joints.

## Figures and Tables

**Figure 1 materials-14-07794-f001:**
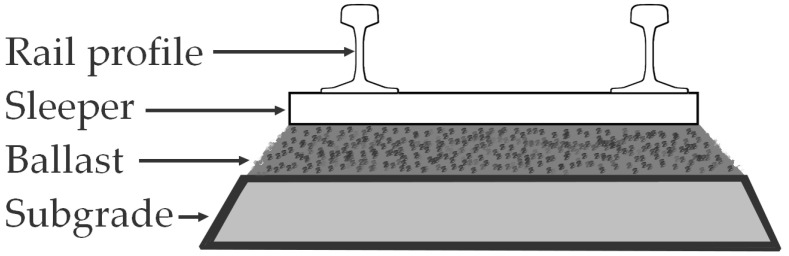
Scheme of railway subgrade elements.

**Figure 2 materials-14-07794-f002:**
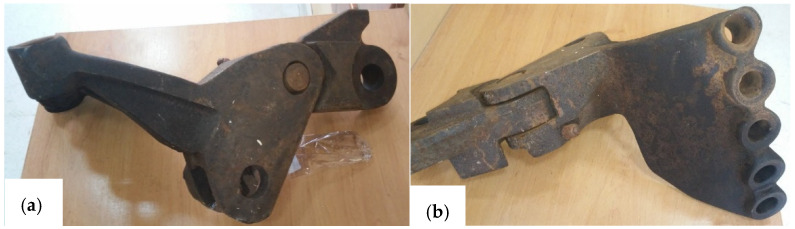
Views of the scraper shovel, linkage and connecting bolts of the RM 80 UHR bed ballast cleaning machine: (**a**) side view, (**b**) top view.

**Figure 3 materials-14-07794-f003:**
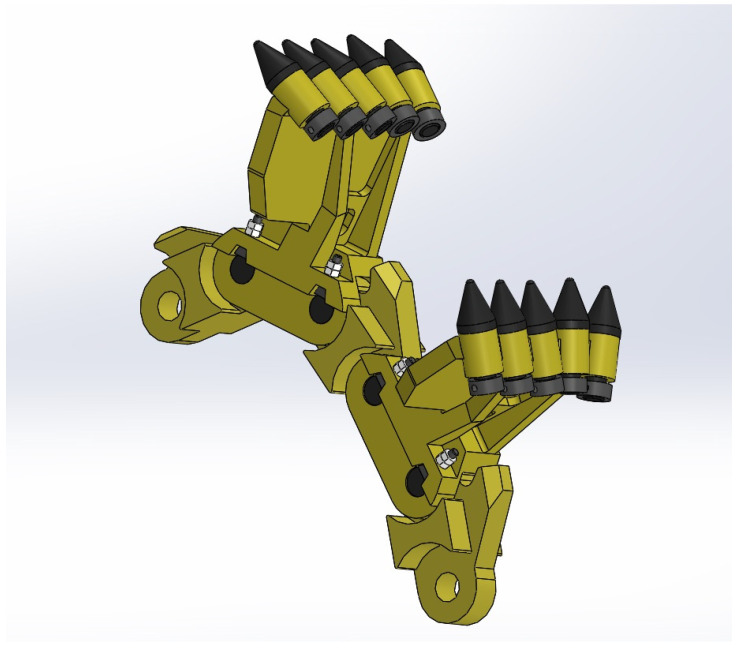
CAD representation of the excavating chain segments, designed in SolidWorks 2018.

**Figure 4 materials-14-07794-f004:**
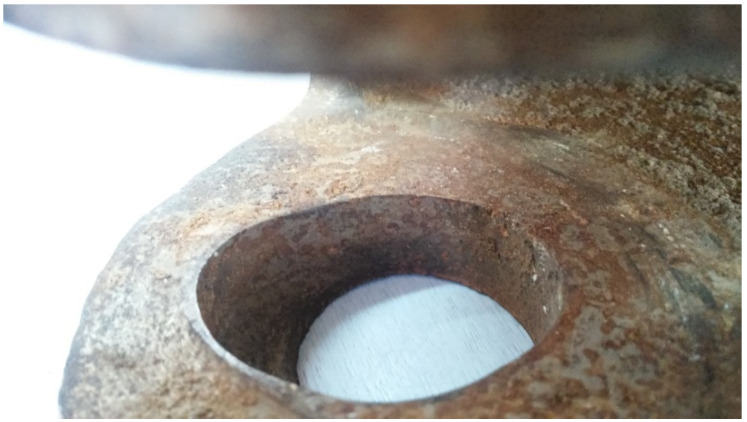
A view on the measured region of the wear-caused dimensional deviations of the inspected parts.

**Figure 5 materials-14-07794-f005:**
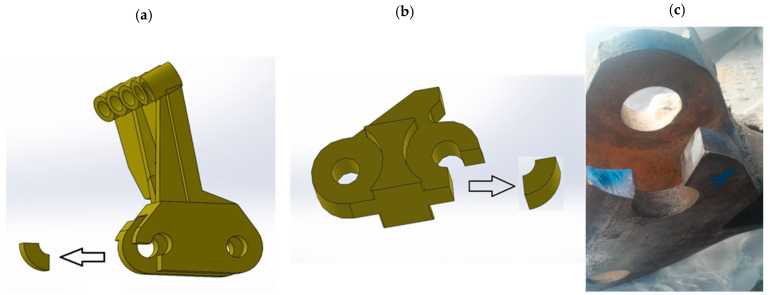
Method of taking samples for testing: (**a**) the scraper shovels, (**b**) the chain connector, (**c**) the photograph of exemplary real-world sample source.

**Figure 6 materials-14-07794-f006:**
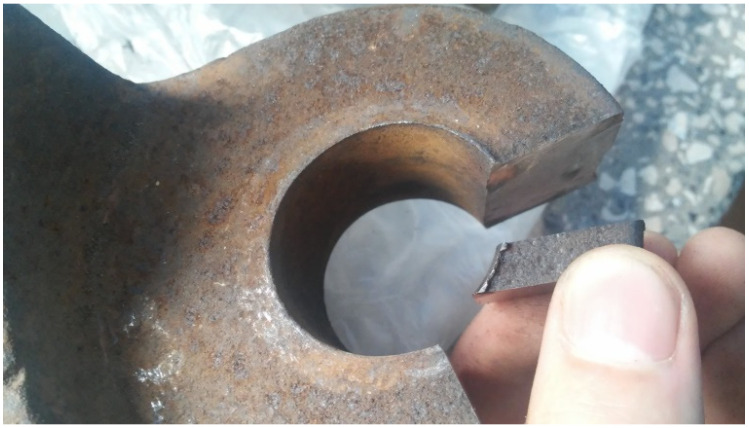
Method of cutting off test samples for metallographic tests.

**Figure 7 materials-14-07794-f007:**
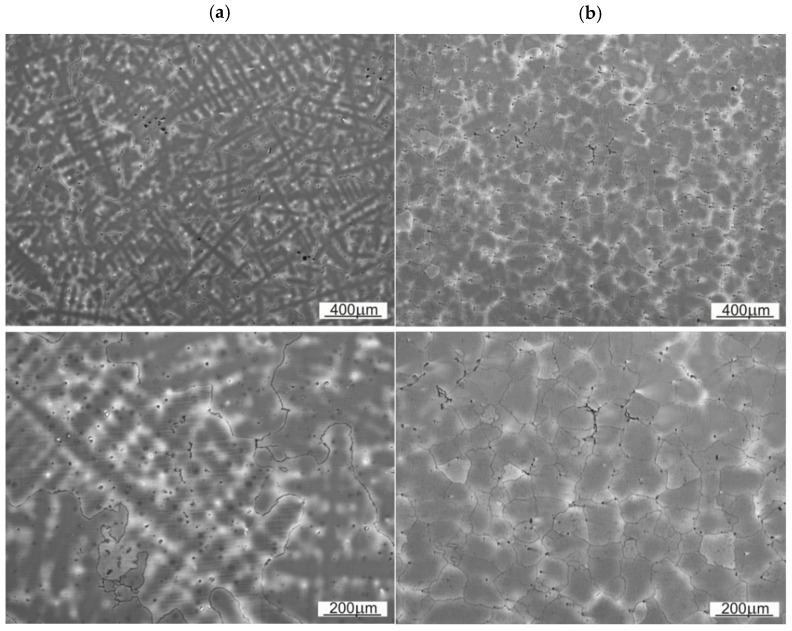
Base material microstructure: (**a**) sample SS1, (**b**) sample SS2.

**Figure 8 materials-14-07794-f008:**
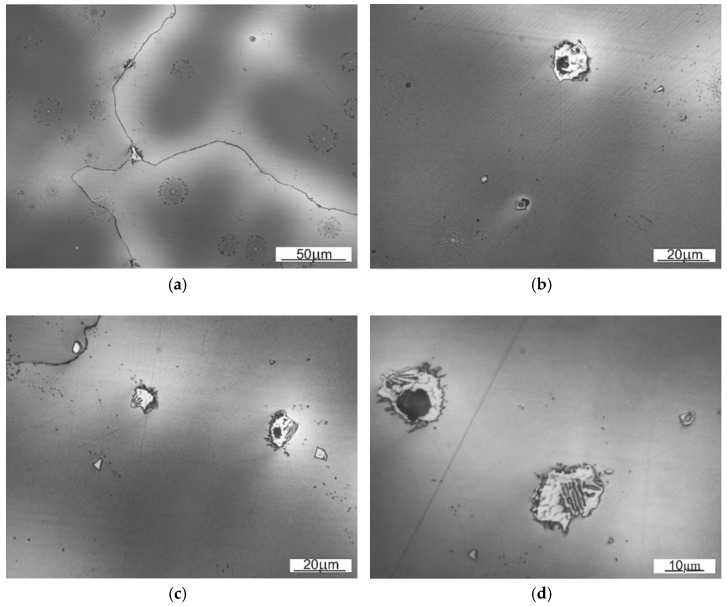
Base material of the sample SS1 with visible carbides and other precipitations: (**a**) 50× magnification, (**b**) 100x magnification, (**c**) 100× magnification, (**d**) 150× magnification.

**Figure 9 materials-14-07794-f009:**
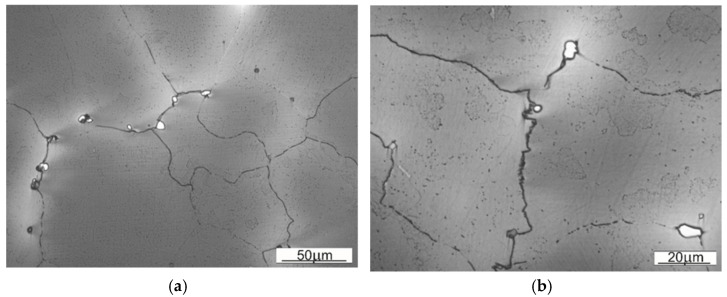
Base material of the sample SS2 with visible carbides and other precipitations. (**a**) 50× magnification, (**b**) 100× magnification, (**c**) 100× magnification, (**d**) 150× magnification.

**Figure 10 materials-14-07794-f010:**
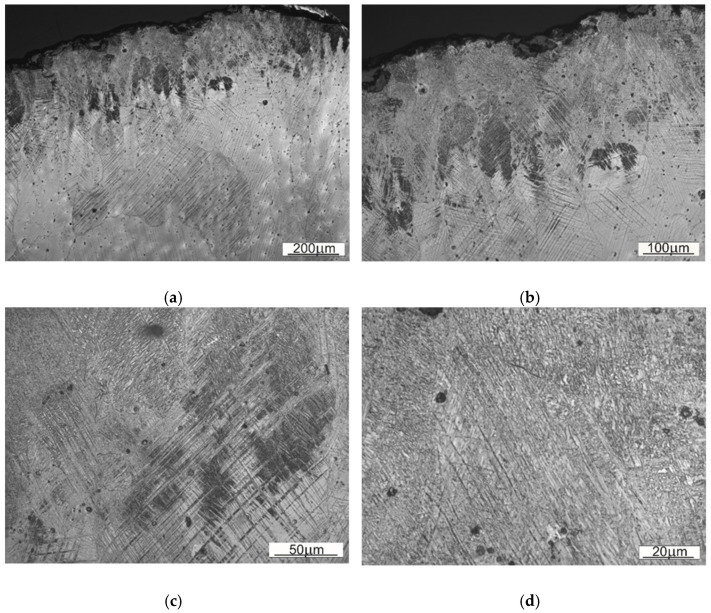
A view on the microstructure of the region located nearby the working surface of the sample SS1: (**a**) 10× magnification, (**b**) 20× magnification, (**c**) 50× magnification, (**d**) 100× magnification, (**e**) 150× magnification, (**f**) 150× magnification.

**Figure 11 materials-14-07794-f011:**
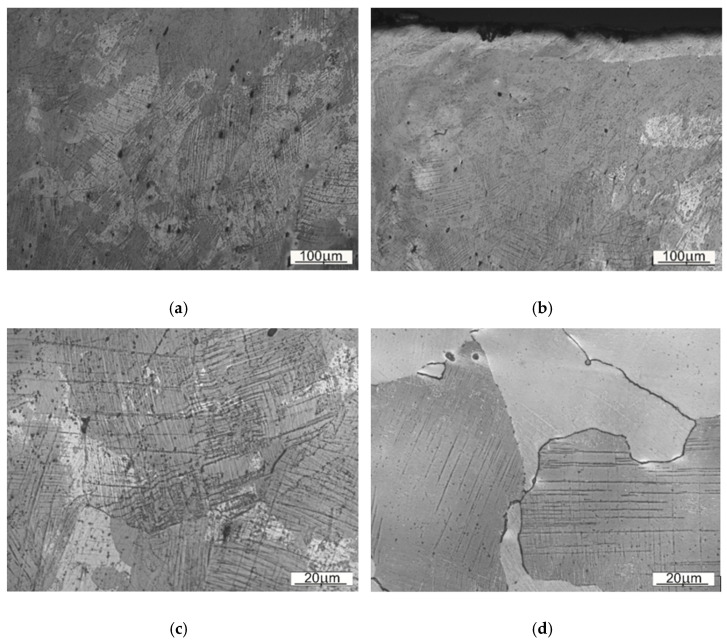
A view on the microstructure of the region located nearby the working surface of the sample SS2: (**a**) 20× magnification, (**b**) 20× magnification, (**c**) 100× magnification (**d**), 100× magnification.

**Figure 12 materials-14-07794-f012:**
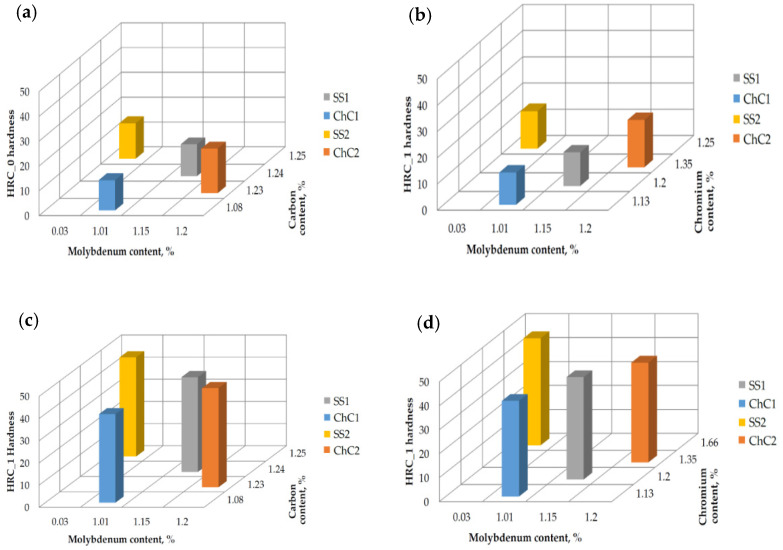
The relationship between the values of HRC_0 and HRC_1 and the content of carbon, chromium and molybdenum content in the particular alloys: (**a**,**b**) HRC_0; (**c**,**d**) HRC_1; (**e**,**f**) increase in HRC.

**Figure 13 materials-14-07794-f013:**
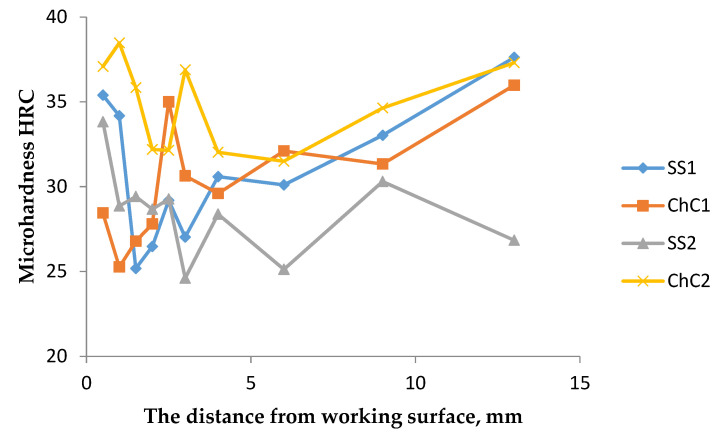
The relationship between HRC microhardness and distance of measurement point from the contact working surface.

**Table 1 materials-14-07794-t001:** Sample enumeration.

Sample Indicator	Description
SS1 (scraper shovels)	scraper shovels material I
ChC1 (chain connector)	chain connector II
SS2 (scraper shovels)	scraper shovels material
ChC2 (chain connector)	chain connector IV

**Table 2 materials-14-07794-t002:** The chemical composition of the investigated parts (optical emission spectroscopy).

Sample No.		Chemical Composition, wt.%							
	C	Si	Mn	P	S	Cr	Mo	Ni	Al
SS1	1.24	0.530	13.0	0.062	0.003	1.20	1.15	0.089	0.023
ChC1	1.08	0.446	12.6	0.046	0.003	1.13	1.01	0.078	0.018
SS2	1.25	0.790	12.4	0.058	0.003	1.66	0.03	0.178	0.049
ChC2	1.23	0.498	12.2	0.057	0.003	1.35	1.20	0.150	0.022

**Table 3 materials-14-07794-t003:** Measurements of dimensional wear in the region of peg hole of the particular samples.

Sample No.	Nominal Diameter, mm	Measured Diameter, mm	Peg Hole Dimensional Deviation, %
SS1	34.2	38.0 ± 5.0	11.1
ChC1		37.0 ± 2.0	8.2
SS2		38.0 ± 4.0	11.1
ChC2		36.5 ± 1.5	6.7

**Table 4 materials-14-07794-t004:** Comparison of average grain size of both materials in samples SS1 and SS2.

Sample No.	Average Grain Size, µm^2^
SS1	49,582
SS2	6409

**Table 5 materials-14-07794-t005:** Values of Brinell hardness for all samples, followed by Rockwell test on raw (HRC_0) and reinforced regions (HRC_1) of inspected materials. The values marked with asterisk are below the scale of Rockwell test, and are given only for comparative purposes.

Sample No.	HBW	HRC_0	HRC_1	Reinforcement, %
SS1	213	12.8 *	42.7	334
ChC1	232	12.2 *	39.9	327
SS2	206	14.3 *	44.7	313
ChC2	237	18.0 *	41.6	231

* conversion value of the HRC scale.

**Table 6 materials-14-07794-t006:** Vickers microhardness values—measured in HV0.5 scale and converted to HRC according to Qvarsnstöm method [51].

Sample No.	Dist. from Working Surf., mm	SS1 HV	HRC	ChC1 HV	HRC	SS2 HV	HRC	ChC2 HV	HRC
1	0.5	291	35.4	333	28.4	347	33.8	362	37.1
2	1.0	269	34.2	294	25.3	336	28.9	375	38.5
3	1.5	280	25.2	298	26.8	269	29.4	351	35.8
4	2.0	287	26.5	293	27.8	278	28.7	320	32.2
5	2.5	343	29.2	297	35.0	297	29.3	319	32.1
6	3.0	308	27.0	265	30.6	281	24.6	360	36.9
7	4.0	300	30.6	291	29.6	307	28.4	318	32.0
8	6.0	319	30.1	269	32.1	304	25.1	314	31.5
9	9.0	313	33.0	305	31.3	327	30.3	340	34.6
10	13.0	352	37.6	280	36.0	367	26.8	364	37.3

## Data Availability

The data presented in this study are available upon request from the corresponding author.

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
