# Peer review of "The Role of Chemical Composition of High-Manganese Cast Steels on Wear of Excavating Chain in Railway Shoulder Bed Ballast Cleaning Machine"

_materials, 2021, doi:10.3390/ma14247794_

Round 1
Reviewer 1 Report
Dear Authors,
I regret to inform you that I cannot give you a positive note on your manuscript. The reasons are listed below. This is not my intention to discourage you in any way at all. This is just my completely objective opinion as a reviewer.
Title: “The role of the chemical composition of high-manganese cast steels on wear of excavating chain in railway shoulder bed ballast cleaning machine”
The manuscript deals with the wear and work hardening of a high manganese cast steel used in ballast excavating chains. The results presented in the manuscript provide valuable information on the damages caused in the chain elements in form of wear and work hardening due to plastic deformation. The level of the information in the manuscript, however, does not meet the minimum criteria to be considered as a scientific contribution worth publishing. It is more a report including hardness measurements, microstructure analysis, and some qualitative wear assessment. What this manuscript is lacking in order to fulfill the conditions of an original scientific article is as follows:
1. The introduction section does not provide any reference on the influence of alloying elements and/or thermo-mechanical treatment, and consequently the size of austenite grains on properties of the studied material.
- The experimental work regarding the material is quite poor; it only includes some images of microstructure, hardness measurements, chemical composition, etc. This information does not provide means of original scientific finding regarding material processing and performance.
- The discussion section continues with the interpretation of the influence of alloying elements on austenite grain size and consequently on the hardenability of the material, which is not a finding from this work only, it is a known fact from the literature, which was not cited! The same goes for the wear behavior of the material as a function of alloying elements, austenite grain size, and hardness.
- Having no original scientific finding, the authors did not provide the Conclusion section at all.
Therefore, as stated above, without any intention to discourage the authors, my suggestion is that this article should not be accepted for publication.
Author Response
Dear Reviewer,
Thank you very much for reviewing our manuscript. According to the comments and the questions, we have carefully revised our paper. Below we would like to provide the answers:
Remark 1
The introduction section does not provide any reference on the influence of alloying elements and/or thermo-mechanical treatment, and consequently the size of austenite grains on properties of the studied material.
Response: Thank you for this notice. The introduction was enriched by additional theoretical study of the studied topic. Appropriate references were also introduced into the manuscript.
Remark 2
The experimental work regarding the material is quite poor; it only includes some images of microstructure, hardness measurements, chemical composition, etc. This information does not provide means of original scientific finding regarding material processing and performance.
Response: Thank you for this notice. We are aware, that our paper might resemble some kind of expertise, nevertheless we believe that in the improved form it brings some novelty into the subject. Firstly, by this time we did not stumble upon analyses of wear in this particular, albeit specific machine – most of the exploitation-related papers covered studies of mining or quarry machinery, where the wear mechanism of differently composed Hadfield cast steels is usually different, as it relies on the impact forces, not continuous, high stress. Secondly we introduced a method of estimation of surface work-hardenability by mixing two approaches of hardness testing and we believe that somebody might find it useful.
We consider it remarkable thatto obtain Hadfield specimens with a mirror-like surface finish for metallographic purposes, yet they have to be polished for a long time with almost no load, since when a load is applied the grains near the polished surface can also dislocate and slide along multitude of available planes and either harden the surface (thus affect the following hardness tests) or even make the small scratches impossible to get rid of, mainly due to the fact, that after dislocation of the grains, the scaffold of hard carbides can release some particles into the polishing environment.
Remark 3
The discussion section continues with the interpretation of the influence of alloying elements on austenite grain size and consequently on the hardenability of the material, which is not a finding from this work only, it is a known fact from the literature, which was not cited! The same goes for the wear behavior of the material as a function of alloying elements, austenite grain size, and hardness.
Response: Once again, thank you for underlining, that we did not emphasize what we actually worked on. Our finding was, that in the specific work conditions of ballast cleaning machine excavation chain the chemical composition of different Mangalloys affects strongly the mechanism of the wear, even though the “net” wear (from the operational point of view) seems to be really similar.
Remark 4
Having no original scientific finding, the authors did not provide the Conclusion section at all.
Response: Thank you very much. The conclusion & discussion section has been added.

Reviewer 2 Report
In this work, the effect of the chemical composition on the physical properties of the ballast excavating chains made of high-manganese steels was investigated. The wear mechanism, work hardening ability and hardness in the cross section areas were discussed. Some important results have been obtained. Still, the necessary modifications or clarifications should be made before the possible acceptance.
(1) The language of this paper is not very good, and should be further improved.
(2) The method to evaluate the the average austenite grain size should be given. Why the unit of average grain size is μm2?
(3) The detailed theoretical analysis should be given based on the experimental findings.
(4) The section conclusions should be added.
Author Response
Dear Reviewer,
Thank you very much for taking the time to read our manuscript once again thoroughly and make recommendations for its correction and improvement. We have read the comments carefully and have responded to all your comments.
Remark 1
The language of this paper is not very good, and should be further improved.
Response: Thank you for this notice. We managed to find several mistakes, usually being a result of redesigning the sentences a few times.
Remark 2
The method to evaluate the average austenite grain size should be given. Why the unit of average grain size is μm2?
Response: Thank you for this notice. Average grain size for samples SS1 and SS2 (unloaded) was determined using the Jeffries planimetric method with Saltykov rectangle correction ant its value is in μm2. We corrected it in the text.
Remark 3
The detailed theoretical analysis should be given based on the experimental findings.
Response: Thank you very much. The theoretical analysis has been added.
Remark 4
The section conclusions should be added.
Response: Thank you very much. The discussion section has been added.

Round 2
Reviewer 1 Report
Dear Authors,
I have performed the second round review of your manuscript. I apologize for being late with the process. Attached please find the review report.
Best regards and good luck.

Author Response
Dear Reviewer,
Thank you very much for taking the time to read our manuscript once again thoroughly and make recommendations for its correction and improvement. We read the comments carefully and improved the article according to your suggestions. We standardized the hardness scale throughout the article and added conclusions regarding the change in hardness across the cross-section of the samples. We also reworked the sections "Conclusion" and "Discussion" according to your feedback. Thank you very much once again.
Reviewer 2 Report
The paper has been improved and proper modifications have been implemented to the article. The current version can be accepted for publication.
Author Response
Dear Reviewer,
Thank you very much for taking the time, reading and recommending the manuscript.